# Radiographic Appearance of the Fore Digit and Carpal Joint in the Mule Foal from Birth to 3 Months of Age

**DOI:** 10.3390/ani13152417

**Published:** 2023-07-26

**Authors:** Irene Nocera, Benedetta Sorvillo, Micaela Sgorbini, Benedetta Aliboni, Simonetta Citi

**Affiliations:** 1Institute of Life Sciences, Sant’Anna School of Advanced Studies, Via Santa Cecilia 3, 56127 Pisa, Italy; 2Department of Veterinary Sciences, Veterinary Teaching Hospital “Mario Modenato”, Via Livornese snc, 56122 Pisa, Italy; benedetta.sorvi@gmail.com (B.S.); micaela.sgorbini@unipi.it (M.S.); benedetta.aliboni@libero.it (B.A.); simonetta.citi@unipi.it (S.C.)

**Keywords:** mule foal, carpus, prematurity, growth plate, radiography, fore digit

## Abstract

**Simple Summary:**

Radiography is important for the diagnosis of prematurity and musculoskeletal abnormalities in foals. Given that no data on the normal radiographic anatomical development of the mule foal are available, we evaluated the radiographic development of the mule fore digit and carpal joint from 0 to 3 months of age. The radiographs highlighted that growth plate closure times differed depending on the fore digit bones, and on whether the carpal bones were immature at birth. Distal phalanx was triangular at birth, palmar processes had developed by one month, and proximal sesamoid bones were trapezoidal by three months. No distal phalanx remodeling was evident within the hoof capsule or variations in the hoof axis with growth, and hoof capsule thickness increased. We identified a baseline for the interpretation of radiological features of the forelimb in the mule foal, and we highlighted specific radiographic parameters that were intermediate between those found in horses and donkeys.

**Abstract:**

Few studies have established the normal radiographic anatomical development of the donkey foal and, to date, no data are available for mules. Our aim was thus to evaluate the radiographic development of the fore digit and carpal joint in the mule foal from 0 to 3 months of age. Ten forelimbs of five healthy full-term mule foals were included. Radiographs of the fore digit lateromedial and dorsopalmar and the carpus dorsopalmar were performed weekly for the first month of age, and bi-monthly during the following two months. Fore digit growth plate closure times, morphological, angular, and linear radiographic parameters, and also carpal cuboidal bone mineralization were evaluated. Growth plates were graded as open, closing, and closed. Carpal bone appearance was graded as mature, slightly immature, or immature. Growth plate closure times showed the following: middle phalangeal distal physis (DP) closed at birth, and proximal physis (PP) started to close at around two months; first phalanx DP closed at seven days, and PP started to close at three months; and third metacarpal bone DP started to close at two months. Carpal bones were immature at birth. Distal phalanx (PD) was triangular at birth, palmar processes had developed by one month, and proximal sesamoid bones were trapezoidal by three months. The hoof wall, PD dorsal wall, and palmar angles values changed in parallel, and hoof capsule thickness increased. No PD remodeling evidence within the hoof capsule or variations in the hoof axis with growth were found. We identified a baseline for the interpretation of forelimb radiological features in mule foals. The specific features found were intermediate between those found in horses and donkeys.

## 1. Introduction

Mules result from the crossing between a male donkey (*Equus asinus*) and a female horse (*Equus caballus*). Interspecies breeding has led to a hybrid offspring with unique chromosomal, physical, and physiological characteristics [1,2]. Due to the genetic variations between the equine species and hybrids, data generated specifically for horses, such as laboratory reference ranges, husbandry practices, and procedural techniques, may not be applicable to equine hybrids [1,2].

Historically, mules were used in agriculture development, transport, and traction [3,4]. They were bred as draft animals throughout Europe for centuries––above all in France and Italy [2]. Mules are faster, stronger, and taller than donkeys and have more endurance than horses, and today they are involved in work, sports, and pleasure riding [5].

Breeding has resulted in clinically relevant differences in the anatomy and physiology of mules compared to donkeys and horses [1,2]. Morphometric and functional measurements have thus been investigated in mules in order to optimize working and sports practices [6,7]. Extensive studies have been performed on the baseline for the radiological interpretation of mature and immature horse digits [8]. Recent studies have investigated donkey limbs, looking for potential differences between breeds [9,10,11], and have highlighted how donkey foal growth plates close at an older age than in horses [11]. Interestingly, the radiographic closure times of the growth plates of the distal extremity of the manus in donkeys have been shown to be considerably later than those for horses [8,12].

Closure times have typically been seen as being dependent both on genetic factors [8,12] and on different exogenous influences [13]. However, to the best of our knowledge, there are only a few studies on radiological baseline findings in mules [5], and none in neonatal mule foals.

In neonatal foals, it is also essential to investigate the grade of ossification of the cuboidal bones, given that poor grades are a common finding in premature and dysmature foals. Radiographs of the carpus and tarsus should be performed in any high-risk foal to obtain a diagnosis. However, there have been few studies on the offspring of mule foals [14,15]. Mules seem to be more resistant, yet mule neonate survival is still a major problem, where high mortality is due to postpartum conditions, such as prematurity [14]. In the case of incomplete cuboidal ossification, the treatment of choice aims to limit weight bearing and exercise, and early treatment is pivotal to prevent relevant orthopedic complications, such as angular limb deformities, degenerative joint disease, and osteochondrosis dissecans [16]. Thus, radiological baseline findings provide clinicians with essential information on the physiological anatomical development and normal closing time of growth plates in the distal limb. However, there are currently no data available for mule foals.

The aim of this study was to evaluate the radiographic appearance of the anatomical development of the fore digit and carpal joint in mule foals from 0 to 3 months of age.

## 2. Materials and Methods

### 2.1. Study Population

A total of 10 forelimbs were evaluated of five full-term mule foals, born at the Veterinary Teaching Hospital of University of Pisa, between 2020 and 2022, were evaluated. Foals were born from cross between Standardbred mare and Amiata donkey stallion. Foals were healthy and musculoskeletal sound based on physical and orthopedic examinations. All foals underwent the same management conditions: after the delivery they were housed with their mares in individual 6 × 6 m boxes until the second week postpartum, then in individual soft ground paddocks. Mares were fed with meadow hay ad libitum along with commercial equine feed, in line with the NRC energy recommendations [17]. Mule foals were able to nurse from their mares throughout the study. Data concerning body weight (BW) were recorded throughout the study.

### 2.2. Radiographic Protocol

Radiographic examinations were performed on forelimbs every week for the first month of age and bi-monthly during the following two months for a total of six radiographic examinations, detailed as follows: T0 (within 24 h after birth), T1 (at 7 days of life), T2 (at 14 days of life), T3 (at 21 days of life), T4 (at 28 days of life), T5 (at 2 months of life), and T6 (at 3 months of life). 

Prior to radiography, the limbs and hooves were cleaned thoroughly. When necessary, the subjects were sedated in order to achieve an accurate positioning for radiographic execution, with detomidine chloride (10 μg/Kg, IV) (Detogesic R, Zoetis Italia, Rome, Italy) and buthorphanol 0.025 mg/kg, IV (Nargesic, ACME, Cavriago, Italy).

Three radiographic views of each forelimb were obtained as follows: lateromedial (LM) and dorsopalmar (DP) of the fore digit, and DP of the carpus [18]. For digit LM and DP views, both feet were positioned in a weight bearing position on a flat wooden block (4 cm high) to provide an even weight distribution between the front limbs [10]. All radiographs were obtained with a focal distance of 80 cm, and the X-ray beam was perpendicular to the X-ray cassette and parallel to the ground [8]. The radiographic images were obtained with a portable X-ray generator (Gierth HF100 M), with exposure factors of 55 kV and 3 mAs for the digit views [11] and 60 kV and 5 mAs for the carpus views [19]. Images were acquired on a flat panel by an “Exam-ion X-DR Portable Case M WiFi DT” system. One single experienced operator performed all the radiographs (I.N.). Radiographs were scanned, and the digitized images were analyzed using commercial software (HorosTM–DICOM, https://horosproject.org/, accessed on 1 June 2023). For digit LM views, the foot and fetlock were included on a single radiographic image, depending on the size and conformation. When these conditions were not possible, two separate views of the hoof and the fetlock were obtained [11].

### 2.3. Radiological Parameters

A series of seven morphological anatomical elements were evaluated in all 10 fore digits in accordance with the literature [9]: PD (*Phalanx distalis*) dorsal margin, PD extensor process (*Processus extensorius)*, PD palmar margin, navicular bone, palmar process (*Processus palmares*), Mc3 sagittal ridge, and proximal sesamoid bones (Figure 1).

The following digit growth plates were evaluated [12]: PM (*Phalanx media*) distal and proximal physis, PP (*Phalanx proximalis*) distal and proximal physis, and Mc3 (Third Metacarpal Bone) distal physis. Physis closure times were evaluated and graded as follows [11]: open (radiolucent), closing (unclear), and closed (radio dense). Carpal bone appearance was assessed and graded as mature, slightly immature, or immature, according to the previous literature [18].

In addition, in radiographs where the angle between the vertical midline of the Mc3 and a perpendicular line to the ground floor was inferior to 15°, a series of 13 angular and linear direct radiometric parameters of the digit were evaluated [11] (Figure 2 and Figure 3). Angular parameters were as follows [11]: hoof angle (HA), dorsal distal angle of PD (DPDS), angle of solar aspect of PD (PA), proximal palmar compact bone angle (PCA), reflex angle of palmar compact bone (AA), distal apex angle PD (APPD), angle between Mc3 and PP (Mc3-PP), angle between PP and PM (PP-PM), and angle between PM and PD (PM-PD). Linear parameters were as follows [11]: founder distance (D), integument depth of the dorsal aspect of the foot, proximal site (IDA), integument depth of the dorsal aspect of the foot (mid-dorsal site) (IDM), integument depth of the dorsal aspect of the foot (distal site) (IDB).

Evaluation of the data was performed by two experienced operators in consensus [11].

### 2.4. Statistical Analysis

The distribution of the angular and linear radiometric parameters was evaluated using the Shapiro–Wilk test. The results were reported according to the distribution as mean and standard deviation (*p* ≥ 0.05), or as median and quartile values (*p* < 0.05). Statistical analysis was carried out using IBM SPSS Statistics v. 27.0 (IBM Corp., Armonk, NY, USA). 

## 3. Results

The median body weight values throughout the study period were as follows: 44 kg (34–52 kg) at T0, 60 kg (40–62 kg) at T1, 72 kg (47–77 kg) at T2, 80 kg (60–83 kg) at T3, 90 (70–90 kg) at T4, 109 kg (87–114 kg) at T5, and 143 kg (125–150 kg) at T6.

A total of 360 radiographs were evaluated. With regard to morphological parameters (Figure 4), the PD was triangular to rounded at birth, and then started to flatten with growth. At birth, the palmar only started to develop at one month of life. The navicular bone was rounded at birth and developed for two months after birth. The proximal sesamoid bones were seen in continuous development from a single center of ossification and acquired a trapezoidal shape at around three months of age.

The results of growth plate closure times showed that distal physis of the PM was closed at birth; PM proximal physis started to close around two months of life; distal physis of the PP appeared as a clear radiolucent line at birth and closed at seven days, and the scar was visible during all of the study period; PP proximal physis started to close at 3 months; and, finally, distal physis of Mc3 started to close at two months (Figure 5).

Carpal bones presented a slightly immature appearance from birth to seven days: the fourth carpal bone was rounded, and there was a large space between the bones (Figure 6).

The results for the angular and linear parameters were assessed on 4 out of 10 fore digits, in which the angle between the vertical midline of the Mc3 and a perpendicular line to the ground floor was inferior to 15, as reported previously [11]. The results were reported in Table 1 and Table 2, respectively. HA and DPDS showed changing values in the first three months of life in a parallel trend (HA: from 66.9° ± 5.4° to 54.6° ± 3.8°; DPDS: from 62.1° ± 3.2° to 51.2° [50.5°–58.3°]). PA had a similar behavior in line with previous angles (from 10.3° ± 6.5° to 3.0° (2.8°–6.0°)), showing that PD palmar border became progressively parallel to the ground.

The PCA, AA, and APPD angles showed no particular variations, with no significant remodeling of PD within the hoof capsule. No notable variations were detected for hoof axis indicators, such as Mc3-PP, PP-PM, and the PM-PD axis. Finally, linear parameters (i.e., D, IDA, IDM, and IDB) showed increasing values in the first three months of life, showing an increase in hoof capsule thickness.

## 4. Discussion

The present study described the radiological anatomy of the distal limb and the carpus of the mule foal from 0 to 3 months of age. To the authors’ best knowledge, this is the first study that described the radiographic development and closure times of the growth plates and morphological appearance of the distal extremity of the manus and cuboidal bone ossification in the mule foal for the baseline interpretation of radiographic data of the mule foal.

The distal phalanx was triangular at birth, the palmar processes were not evident at birth, and the navicular bone presented an oval outline evolving in a trapezoidal shape, in agreement with previous morphological descriptions of the horse and donkey foal [8,11]. The extensor process continued osseous development until two months of age, similar for horses (1–3 months) and donkeys (4.5 months) [8,11]. Finally, the proximal sesamoid bones showed a single ossification center, developing from a triangular-rounded shape to trapezoidal, as in donkeys. A small percentage of horses might be ossified from two ossification centers, which was not observed in our population [8,11].

Our findings show that growth plates of the digit presented intermediate features and different anatomical development compared to horse and donkey foals. In our study, PM distal physis was closed at birth, as in the horse, but in donkey foal, it closes in the first four days of life [8,11]. PM proximal physis was closing at two months of life, whereas in donkeys, it closes at three months of life. Data on horses reported the closure at eight months [8,11]. In our mules, PP distal physis was closed until one week of life, as for donkeys, but different from horses, in which it should be completely closed at birth, and any lucent line might be signify prematurity. In our study, PP proximal physis closed at three months in mules, whereas in donkeys it closes at 4.5 months, and in horses, it is completely closed at twelve months [8,11]. Finally, Mc3 distal physis was closed at two months of life, unlike horses, for which it closes at three months, and for donkeys, it closes at three and a half months [8,11]. In the literature, the closure growth plates have been reported radiographically in different horse breeds along with the average estimate of months of physical closure for anatomical sites in the limb [19], but there is no specific information on fore digit closure in Italian standardbred. Moreover, to the best of the authors’ knowledge, there is no specific information in the literature regarding differences between donkey breeds. However, variations in the closure of the growing plates may be related to breed.

On evaluation of carpus bone ossification, the mule foal showed rounded margins for cuboidal bones and a wide space between them. These findings differ from horses in which the margins of the cuboidal bones are straight with square corners, and there is very little space between them [8,18]. On the other hand, no data were available for donkeys. Ossification of the cartilaginous templates of the cuboidal bones begins at the center and gradually extends to the periphery due to endo-chondral ossification [16]. Thus, in a full-term foal, the ossification extends to the periphery of the cartilaginous template, while in the premature or dysmature foal, ossification can be delayed, and this is most notable at the periphery because this is the last area to become ossified [16]. In the mule foal, the immature appearance of the cuboidal carpus bone might lead to mistaken evaluation of the ossification extension for these bones. However, the present radiological baseline findings might help the clinician provide essential information to properly assess neonatal radiographic images of a carpus.

With regards to angular and linear parameters, our findings showed that HA, DPDS, and PA values gradually decreased in parallel with age. Donkey foals showed similar mean angle values at 0–1 year [11]. We found similar values for PCA, AA, and APPD in the first three months of life, while donkey foals showed that the palmar compact bone border of the PD was becoming increasingly vertical with age (from 0 to 2 years of age) [11]. In our study, hoof axis indicators were similar over time, and these results were similar to donkey foals for PP-PM angle, but the angle between Mc3 and PP and between PM and PD decreased with age [11,20]. Finally, hoof thickness values tended to progressively increase till three months of life; however, no evidence of correlation with age was detected in donkey foals, which generally showed lower values compared to mule foals [11,21].

In the present study, the small sample size might have influenced our results, and may thus represent a limitation. The study group was homogeneous for the crossbred mule in relation to both housing and management, and, therefore, potential influences of these variabilities were not evaluated. Studies on morphological features of limbs in donkeys and horses have been conducted on both heterogenous and homogenous groups, providing data on the differences between the breeds and body size [10,11] as well as management conditions [16,18]. Our study contributes by collecting what we believe to be the first data on healthy mule foals. The potential differences in the closure time of growth plates according to gender were not evaluated due to the limited number of limbs enrolled in our study. However, there are contrasting findings in the literature. Few studies reported that the sex had no significant effect on closure time [22,23], and other studies have shown earlier closure time in females, both in horses [24] and in donkeys [11]. In our study, only the forelimbs were assessed for the follow-up description of the radiological anatomy of the distal limb, which might represent a limitation. However, a previous study showed no growing differences between front and hind digits in horses [12]. Further studies on the hock and stifle joints would of interest in providing information on the growth plates and morphology of the cuboidal bones and normal anatomy, since these joints are often affected in case of incomplete ossification [25,26], severe congenital deformity [27,28], and septic joint [29,30] in neonatal foals.

## 5. Conclusions

Our results represent a baseline for the interpretation of radiological features of the forelimb in the mule foal. We have identified certain radiographic parameters, which showed that mule foals have intermediate features between horses and donkeys. Further research is needed to confirm these findings. This baseline radiographic interpretation could be of clinical interest in the radiological examination of mule foals, such as when assessing lameness or as an aid to the corrective trimming and orthopedic correction procedure of young foals with angular and flexural development diseases.

## Figures and Tables

**Figure 1 animals-13-02417-f001:**
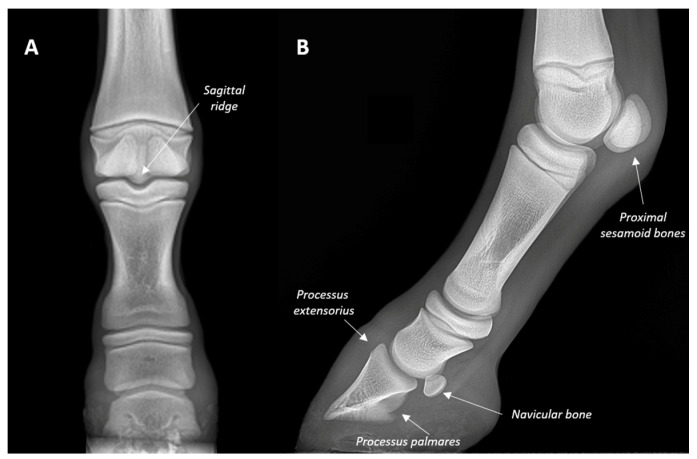
Radiographs of a fore digit showing the morphometric parameters in a T1 mule foal. (**A**) Dorsal-palmar view (lateral is to the left of image and medial is to the right). (**B**) Latero-medial view (dorsal is to the left of image and palmar is to the right).

**Figure 2 animals-13-02417-f002:**
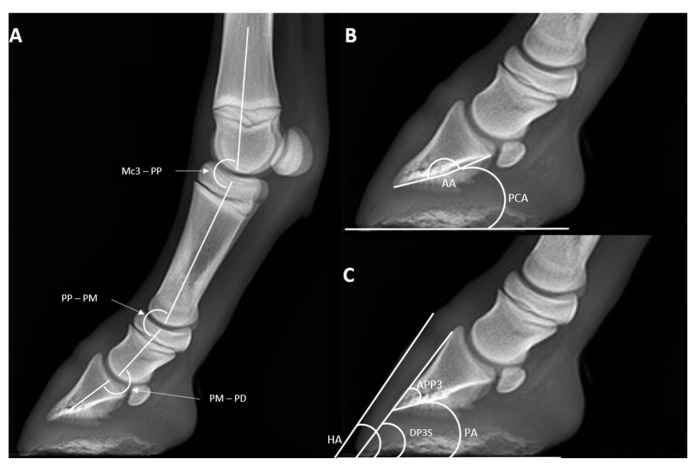
LM radiographs of a fore digit showing the angular parameters in a T3 mule foal. The following angles are indicated: (**A**) Mc3-PP, angle between third metacarpal bone and first phalanx; PP-PM, angle between proximal and middle phalanx; PM-PD, angle between middle and distal phalanx. (**B**) PCA, proximal palmar compact bone angle; AA, reflex angle of palmar compact bone; (**C**) HA, hoof angle; DPDS, dorsal distal angle of PD; PA, angle of solar aspect of PD; APPD, distal apex angle PD. Latero-medial view P––dorsal is to the left of image and palmar is to the right.

**Figure 3 animals-13-02417-f003:**
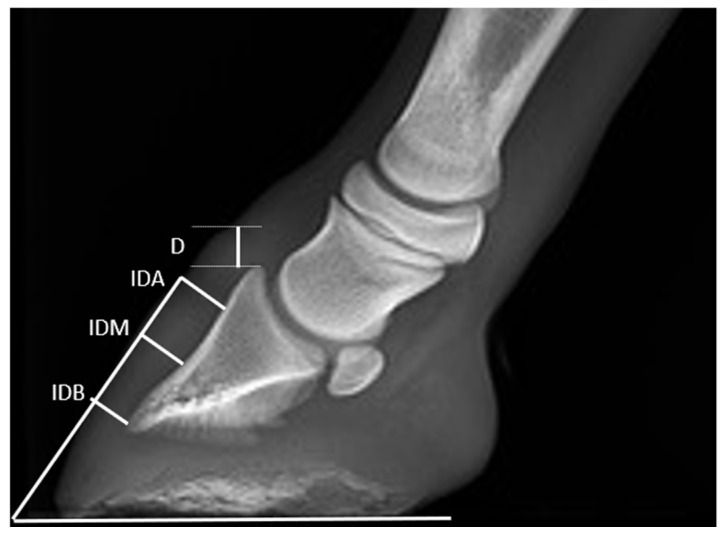
LM radiographs of a fore digit showing the linear parameters, in a T3 mule foal. Key: D, founder distance; IDA, integument depth (proximal site); IDM, integument depth (mid-dorsal site); IDB, integument depth (distal site). Latero-medial view––dorsal is to the left of image and palmar is to the right.

**Figure 4 animals-13-02417-f004:**
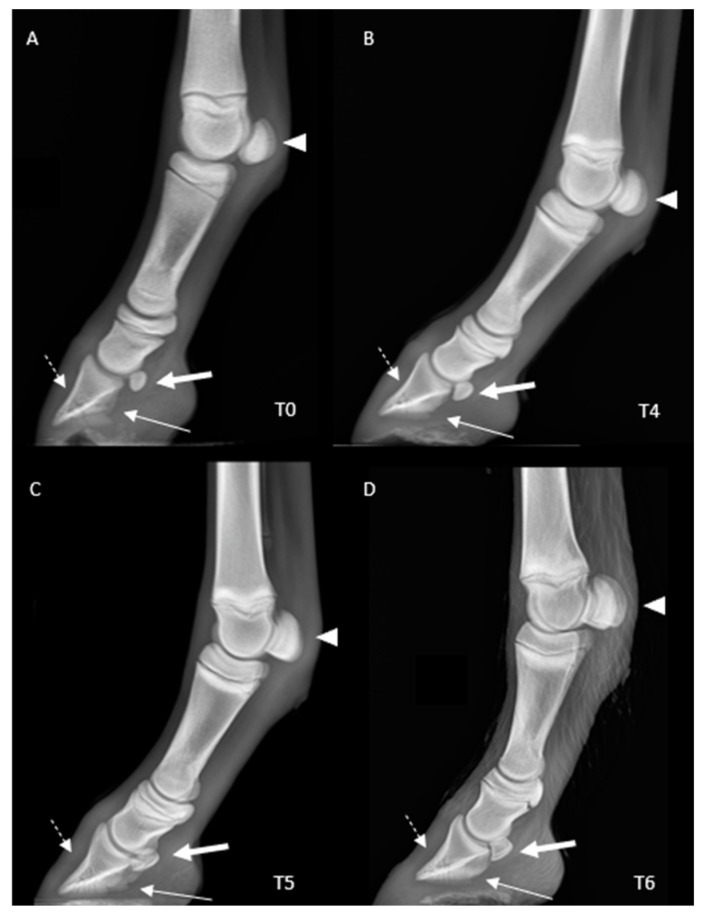
Chronological development of the morphological appearance of the fore digit in a mule foal: 24 h (T0) (**A**), 1 month (T4) (**B**), 2 months (T5) (**C**), and 3 months (T6) (**D**) after birth. Note: the PD is triangular to rounded at birth and flattens with growth (interrupted white arrow). The palmar processes start to develop by one month of life (white arrow); the navicular bone is round at birth and is increasingly developed (double white arrow). The proximal sesamoid bones develop from a single center of ossification and acquire a trapezoidal shape by 3 months of age (arrowhead). Key: PD: Distal Phalanx. Latero-medial views––dorsal is to the left.

**Figure 5 animals-13-02417-f005:**
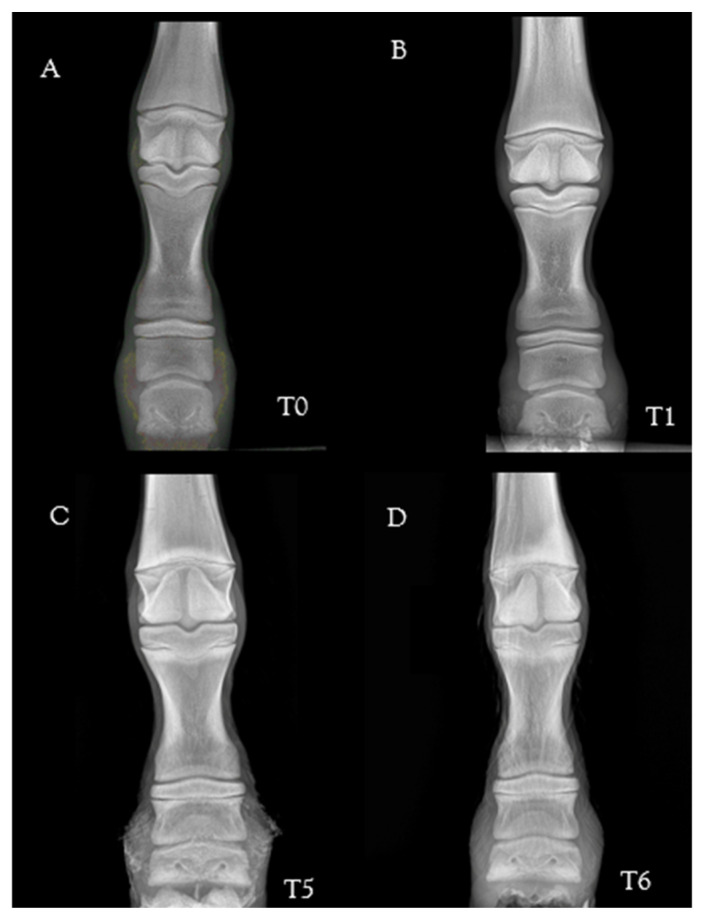
Radiographic appearance of the fore digit growth plates over time in a mule foal: 24 h (T0) (**A**), 7 days (T1) (**B**), 2 months (T5) (**C**), and 3 months (T6) (**D**) after birth. Note: distal physis of the PM is closed at birth; PM proximal physis starts to close by two months of life; distal physis of the PP appeared as a clear radiolucent line at birth and closed at seven days; PP proximal physis starts to close at three months; and distal physis of Mc3 starts to close at two months. Key: PM: Middle Phalanx; PP: Proximal Phalanx; Mc3: Third metacarpal bone. Dorso-palmar views––lateral is to the left.

**Figure 6 animals-13-02417-f006:**
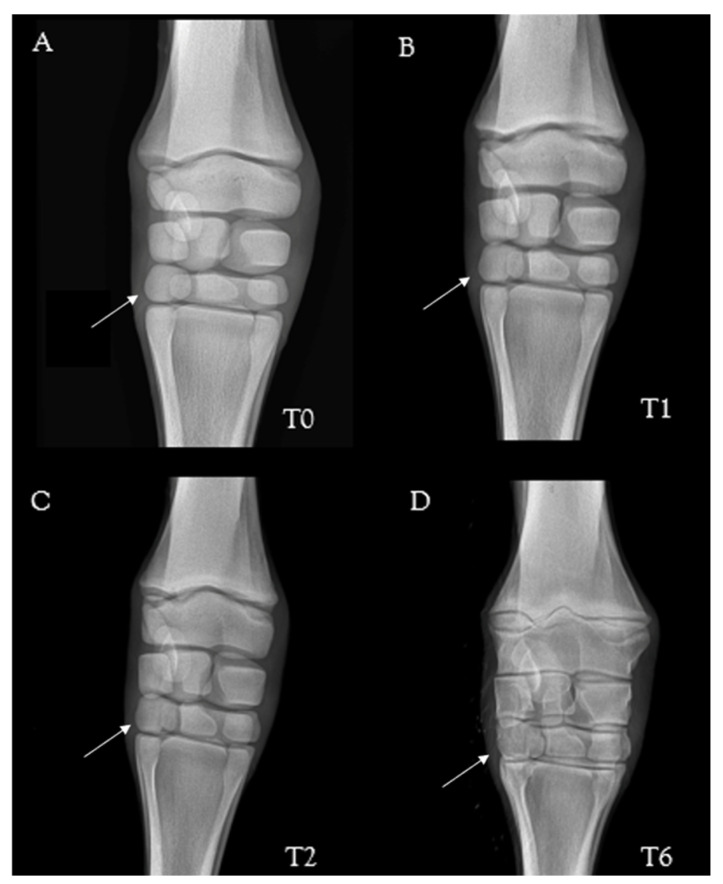
Radiographic appearance of the carpus over time in a mule foal: 24 h (T0) (**A**), 7 days (T1) (**B**), 14 days (T2) (**C**), and 3 months (T6) (**D**) after birth. Note: the fourth carpal bone is particularly rounded at birth (T0) (white arrow). Dorso-palmar views––lateral is to the left.

**Table 1 animals-13-02417-t001:** Digit angular parameters results (expressed in degree). Values were expressed according to mean and standard deviations, except for T2 APPD, T2 Mc3-PP, T6 DPDS, and T6 PA, expressed as median value and quartile values.

	HA	DPDS	PA	PCA	AA	APPD	Mc3-PP	PP-PM	PM-PD
**T0**	66.9 ± 5.4	62.1 ± 3.2	10.3 ± 6.5	33.6 ± 2.1	169.9 ± 2.5	38.3 ± 2.9	164.1 ± 2.4	169.9 ± 2.3	178.4 ± 10.7
** *p* ** **-value**	0.54	0.35	0.25	0.71	0.99	0.80	0.72	0.63	0.97
**T1**	64.6 ± 4.1	57.9 ± 5.6	4.9 ± 4.3	28.0 ± 9.6	169.1 ± 6.9	36.7 ± 2.4	156.3 ± 8.0	166.8 ± 4.1	179.7 ± 4.1
** *p* ** **-value**	0.33	0.31	0.17	0.11	0.25	0.29	0.62	0.11	0.67
**T2**	61.9 ± 5.1	55.5 ± 2.9	6.9 ± 2.0	24.2 ± 4.8	169.1 ± 5.2	36.3 (36.0–39.5)	156.6 (147.9–165.3)	165.8 ± 5.2	181.9 ± 5.6
** *p* ** **-value**	0.45	0.74	0.13	0.63	0.41	0.01	0.03	0.36	0.75
**T3**	60.2 ± 1.6	53.7 ± 2.1	5.5 ± 3.7	23.5 ± 6.2	171.4 ± 6.4	35.9 ± 1.7	155.0 ± 3.4	163.1 ± 1.9	177.7 ± 7.3
** *p* ** **-value**	0.83	0.40	0.07	0.56	0.35	0.11	0.71	0.30	0.73
**T4**	64.3 ± 1.8	52.9 ± 3.7	4.7 ± 0.8	24.6 ± 6.3	169.1 ± 4.0	36.8 ± 1.4	153.9 ± 1.3	162.6 ± 2.9	181.5 ± 3.3
** *p* ** **-value**	0.63	0.76	0.94	0.50	0.20	0.70	0.96	0.87	0.58
**T5**	57.6 ± 2.9	53.3 ± 3.6	3.2 ± 2.2	23.9 ± 5.8	171.5 ± 5.1	37.8 ± 1.7	158.2 ± 2.6	161.9 ± 2.0	177.9 ± 7.8
** *p* ** **-value**	0.10	0.91	0.29	0.24	0.99	0.22	0.23	0.80	0.58
**T6**	54.6 ± 3.8	51.2 (50.5–58.3)	3.0 (2.8–6.0)	25.7 ± 10.9	170.8 ± 4.7	36.6 ± 1.8	163.8 ± 2.3	164.7 ± 4.8	181.4 ± 13.8
** *p* ** **-value**	0.43	0.02	0.01	0.29	0.86	0.07	0.90	0.09	0.12

**Table 2 animals-13-02417-t002:** Digit linear parameters results (expressed in mm). Values were expressed according to mean and standard deviations, except for T1 IDA, T2 IDA, T2 IDB, T3 D, and T5 IDB, expressed as median value and quartile values.

	D	IDA	IDM	IDB
**T0**	8.8 ± 1.7	9.8 ± 1.1	10.0 ± 1.2	6.6 ± 0.5
** *p* ** **-value**	0.95	0.86	0.17	0.47
**T1**	7.4 ± 1.4	10.5 (10–11)	10.6 ± 1.1	6.9 ± 0.8
** *p* ** **-value**	0.83	0.02	0.60	0.57
**T2**	7.2 ± 1.6	11 (10.1–11)	10.5 ± 1.2	7 (7–7.2)
** *p* ** **-value**	0.54	0.001	0.55	0.001
**T3**	6.6 (6.2–7)	11.0 ± 1.4	10.8 ± 1.1	7.5 ± 0.5
** *p* ** **-value**	0.02	0.16	0.50	0.12
**T4**	6.7 ± 1.1	11.5 ± 1.3	10.7 ± 1.9	8.8 ± 1.7
** *p* ** **-value**	0.72	0.97	0.08	0.85
**T5**	7.8 ± 0.9	12.0 ± 1.4	11.8 ± 1.5	9.5 (9–10)
** *p* ** **-value**	0.27	0.16	0.22	0.02
**T6**	9.0 ± 0.8	12.5 ± 1.7	13.0 ± 0.8	11.3 ± 1.5
** *p* ** **-value**	0.68	0.19	0.68	0.22

## Data Availability

The data presented in this study are available on request from the corresponding author.

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
