# Peer review of "Radiographic Appearance of the Fore Digit and Carpal Joint in the Mule Foal from Birth to 3 Months of Age"

_animals, 2023, doi:10.3390/ani13152417_

Round 1
Reviewer 1 Report
This report addresses the radiographic appearance of the digit and carpus in neonatal mules to 3 months of age. The topic is interesting for equine clinicians as orthopedics problems are commonly seen in equine neonates and we don’t have information regarding the normal radiographic appearance in mules.
The length and structure are correct, and in general terms, it is effortless to understand.
However, there are a couple of questions that should be explained before publication.
It could be interested to include tarsus and stifle to have information regarding growth plates and morphology of the cuboidal bones. Moreover, septic joints more frequently affected are hocks and this kind of studies can help to determinate the degree of bone affectation.
Materials and Methods
2.1. Study population. Could the authors specify the breed of the mares used in the study? There are articles showing differences in the closure of the growing plates regarding the breed and this fact could have influenced in the results.
Animals (Basel). 2021 Dec; 11(12): 3402.
Published online 2021 Nov 29. doi: 10.3390/ani11123402
Growth and Bone Development in the Horse: When Is a Horse Skeletally Mature?
Chris W. Rogers,1,2,* Erica K. Gee,1 and Keren E. Dittmer1
2.2. Radiographic protocol. Was it not needed sedation? Probably in the neonatal mules it was possible to perform the X Rays without sedation, but in the 3 months old seems difficult positioned without help.
Results. Authors have included a Shapiro-Wilk test with statistical significance of p≤0.05. However, there is not any table o data showing these results. Authors only included tables with the descriptive statistical. Could the author provide the data results and explain if there was any statistically significant difference.
The limitations should include the small sample and the lack of data from the tarsus and stifle. It is true that there are no growing plates differences between front and hind digits according to previous articles, but this is different for the rest of the hind limb structures.
Author Response
Revisions were highlighted in green
This report addresses the radiographic appearance of the digit and carpus in neonatal mules to 3 months of age. The topic is interesting for equine clinicians as orthopedics problems are commonly seen in equine neonates and we don’t have information regarding the normal radiographic appearance in mules. The length and structure are correct, and in general terms, it is effortless to understand. However, there are a couple of questions that should be explained before publication. It could be interested to include tarsus and stifle to have information regarding growth plates and morphology of the cuboidal bones. Moreover, septic joints more frequently affected are hocks and this kind of studies can help to determinate the degree of bone affectation.
Materials and Methods
2.1. Study population. Could the authors specify the breed of the mares used in the study? There are articles showing differences in the closure of the growing plates regarding the breed and this fact could have influenced in the results.
Answer: thank you for your suggestion. Details have been added to the main text and the reference has been added (please see “2.1 Study population” and “Discussion” and “References”)
Animals (Basel). 2021 Dec; 11(12): 3402.
Published online 2021 Nov 29. doi: 10.3390/ani11123402
Growth and Bone Development in the Horse: When Is a Horse Skeletally Mature?
2.2. Radiographic protocol. Was it not needed sedation? Probably in the neonatal mules it was possible to perform the X Rays without sedation, but in the 3 months old seems difficult positioned without help.
Answer: thank you for your observation. Details have been added to the main text (please see “2.2 Radiographic protocol”)
Results. Authors have included a Shapiro-Wilk test with statistical significance of p≤0.05. However, there is not any table o data showing these results. Authors only included tables with the descriptive statistical. Could the author provide the data results and explain if there was any statistically significant difference.
Answer: thank you for your observation. The p-vales have been added to table 1 and 2 and statistical analysis paragraph has been updated (please see “2.4 Statistical analysis” and “Results”)
The limitations should include the small sample and the lack of data from the tarsus and stifle. It is true that there are no growing plates differences between front and hind digits according to previous articles, but this is different for the rest of the hind limb structures.
Answer: thank you for your suggestion. Discussion on study limitations has been revised and expanded (please see “Discussion”)
Reviewer 2 Report
The aim of this study was to radiographically evaluate the development of the bones of the finger and wrist of the mule. This is an interesting objective, especially since most of the equine studies are of course concerned with Equus caballus. The material is suitable and the equipment used is appropriate. As movement is essential for the formation of the locomotor organ, please state how the foals were maintained. Were the foals weighed during the experiment? If so, please indicate the growth rate. Please indicate how the foals were fed - this is important for their development. The authors should standardise the nomenclature they use.
The drawings are of high quality. The literature is appropriate.
[78] Please specify what type of exercise they had, i.e. whether they were in the paddock or in the stable and were walked.
[96] x-ray- X-ray
[198] Please change from Distalis to distalis
[117] Please change from Medialis to medialis
[118] Please change from Proximalis to proximalis.
[123] Please change MC3 to Mc3
[175) Please change MC3 to Mc3
[186] Please change MC3 to Mc3
[187} Please change MC3 to Mc3
[187] Please change third Metacarpal bone to Third metacarpal Bone
[208] Please change MC3 to Mc3
[212] Please change MC3 to Mc3
[249] Please change MC3 to Mc3
[271] Please change MC3 to Mc3
Author Response
Revisions were highlighted in red
The aim of this study was to radiographically evaluate the development of the bones of the finger and wrist of the mule. This is an interesting objective, especially since most of the equine studies are of course concerned with Equus caballus. The material is suitable and the equipment used is appropriate.
As movement is essential for the formation of the locomotor organ, please state how the foals were maintained.
Answer: thank you for your suggestion. Details have been added to the main text (please see “2.1 Study population”)
Were the foals weighed during the experiment? If so, please indicate the growth rate.
Answer: thank you for your observation. Details have been added to the main text (please see “2.1 Study population” and “Results”)
Please indicate how the foals were fed - this is important for their development.
Answer: thank you for your suggestion. Details have been added to the main text (please see “2.1 Study population”)
The authors should standardise the nomenclature they use.
Answer: thank you for your suggestion. Revision of the nomenclature has been done through to the main text.
The drawings are of high quality. The literature is appropriate.
[78] Please specify what type of exercise they had, i.e. whether they were in the paddock or in the stable and were walked.
Answer: thank you for your suggestion. Details have been added to the main text (please see “2.1 Study population”)
[96] x-ray- X-ray
Answer: corrected as suggested.
[198] Please change from Distalis to distalis
Answer: corrected as suggested.
[117] Please change from Medialis to medialis
Answer: corrected as suggested.
[118] Please change from Proximalis to proximalis.
Answer: corrected as suggested.
[123] Please change MC3 to Mc3
Answer: corrected as suggested.
[175) Please change MC3 to Mc3
Answer: corrected as suggested.
[186] Please change MC3 to Mc3
Answer: corrected as suggested.
[187} Please change MC3 to Mc3
Answer: corrected as suggested.
[187] Please change third Metacarpal bone to Third metacarpal Bone
Answer: corrected as suggested.
[208] Please change MC3 to Mc3
Answer: corrected as suggested.
[212] Please change MC3 to Mc3
Answer: corrected as suggested.
[249] Please change MC3 to Mc3
Answer: corrected as suggested.
[271] Please change MC3 to Mc3
Answer: corrected as suggested.
Reviewer 3 Report
Dear Authors
The manuscript submitted for review is interesting, but needs additions and corrections.
1. I would complete in the "Introduction" section the information about this one of the most common interspecies hybrids-mules. About the formation of mules, were they the result of natural selection, chance, or conscious work of man? Certainly we know , that already in antiquity crossed horse mares with male donkeys, the hybrid thus created, in terms of external appearance is similar to the mother, and from the father inherits the disposition. Hybridization between species belonging to the same family, leads to the formation of individuals most often infertile. The question arises -If hybrids are not viable from an evolutionary point of view, since they are not fertile, how would they affect evolution? The answer is to be found in genomics; hybrids surprisingly often help the species from which they are descended to adapt better by taking over useful genes from close cousins.
2. The analyzed study group is not a reliable source of knowledge and a sufficient basis for conducting this type of research and drawing conclusions: very laconically given information regarding the study group: 5 individuals (only!), born in the same hospital of the same University. The study, in order to be authoritative, should be carried out in different centers, preferably in different localities, and preferably, for example, in 2 different countries: this gives the probability that these animals may have had different conditions of maintenance, nutrition, health care...etc.
3. I would ask for consistent use of NAV throughout the manuscript (e.g. phalanx distalis, once in lowercase once in uppercase), besides, e.g. the symbol P1-P3 is commonly used in the literature to designate premolar teeth, the symbol "Ph" is used to designate phalanges. "PCA" is commonly appropriate abbreviation for Principal Component Analysis, perhaps another abbreviation can be established in your research?
4) Statistical analyses were performed for too small a study group.
5. Only 21 literature items were cited in the discussion, and this is not a unique topic.
Regards
Minor editing of English language required.
Author Response
Revisions were highlighted in blue.
Dear Authors
The manuscript submitted for review is interesting, but needs additions and corrections.
1. I would complete in the "Introduction" section the information about this one of the most common interspecies hybrids-mules. About the formation of mules, were they the result of natural selection, chance, or conscious work of man? Certainly we know , that already in antiquity crossed horse mares with male donkeys, the hybrid thus created, in terms of external appearance is similar to the mother, and from the father inherits the disposition. Hybridization between species belonging to the same family, leads to the formation of individuals most often infertile. The question arises -If hybrids are not viable from an evolutionary point of view, since they are not fertile, how would they affect evolution? The answer is to be found in genomics; hybrids surprisingly often help the species from which they are descended to adapt better by taking over useful genes from close cousins.
Answer: thank you for your observation. the introduction has been revised and details have been added to the main text (please see “Introduction”).
The analyzed study group is not a reliable source of knowledge and a sufficient basis for conducting this type of research and drawing conclusions: very laconically given information regarding the study group: 5 individuals (only!), born in the same hospital of the same University. The study, in order to be authoritative, should be carried out in different centers, preferably in different localities, and preferably, for example, in 2 different countries: this gives the probability that these animals may have had different conditions of maintenance, nutrition, health care...etc.
Answer: thank you for your observation. These have been added as limitation of the present study and discussed (please see “Discussion”).
3. I would ask for consistent use of NAV throughout the manuscript (e.g. phalanx distalis, once in lowercase once in uppercase), besides, e.g. the symbol P1-P3 is commonly used in the literature to designate premolar teeth, the symbol "Ph" is used to designate phalanges. "PCA" is commonly appropriate abbreviation for Principal Component Analysis, perhaps another abbreviation can be established in your research?
Answer: thank you for your observation. The authors standardise the nomenclature they use through the main text, moreover all the abbreviations used int the paper refer to standardized nomenclature previously reported in literature, and appropriate references were added to the main text (please see “material and methods”).
4) Statistical analyses were performed for too small a study group.
Answer: thank you for your observation. It was performed a descriptive statistical analysis and further details have been descripted in statistical analysis and results paragraphs (please see “2.4 Statistical analysis” and “Results”).
Only 21 literature items were cited in the discussion, and this is not a unique topic.
Answer: thank you for your observation. the main text have been extensively revised and further appropriate references have been added (please see “References”)
Regards
Round 2
Reviewer 2 Report
The authors have made amendments that improve the quality of the work considerably. The work is interesting and has great cognitive and practical value and should therefore be published.
Author Response
.
Reviewer 3 Report
The authors have made most of the recommended improvements.
Author Response
.